Design and optimization of dynamic reliability-driven order allocation and inventory management decision model

Zhang Qiansha 1
Lu Dandan 1 ludandan@gxufe.edu.cn
Xiang Qiuhua 1 2020110004@gxufe.edu.cn
Lo Wei 1
Lin Yulian 2
1 School of Business Administration, Guangxi University of Finance and Economics , Xixiangtang District, Nanning, Guangxi , China
2 School of Siam University Phasicharoen , Bangkok , Thailand
Asif Muhammad
Electronic publication date: 2024 Sep 13
Publication date: 2024
Volume: 10
Electronic Location ID: e2294
Received 2024 Jun 3; Accepted 2024 Aug 9
Copyright: © 2024 Zhang et al.
Copyright year: 2024
Copyright holder: Zhang et al.
License: This is an open access article distributed under the terms of the Creative Commons Attribution License, which permits unrestricted use, distribution, reproduction and adaptation in any medium and for any purpose provided that it is properly attributed. For attribution, the original author(s), title, publication source (PeerJ Computer Science) and either DOI or URL of the article must be cited.
License URL: https://creativecommons.org/licenses/by/4.0/

Keywords: Optimization, Order allocation, Supply chain, Inventory management, Decision support, Reliability

Funding: National Natural Science Foundation of China 72361002 Guangxi Philosophy and Social Sciences Research Project 23AGL001 Guangxi First-class Discipline Applied Economics Construction Project and Guangxi Big Data Analysis of Taxation Research Center of Engineering Construction Projects Fund 2023GSXKB04 This work was supported by the National Natural Science Foundation of China under grant No. 72361002, Guangxi Philosophy and Social Sciences Research Project under grant No. 23AGL001, and Guangxi First-class Discipline Applied Economics Construction Project and Guangxi Big Data Analysis of Taxation Research Center of Engineering Construction Projects Fund under grant No. 2023GSXKB04. The funders had no role in study design, data collection and analysis, decision to publish, or preparation of the manuscript.

==============================
Efficient order allocation and inventory management are essential for the success of supply chain operations in today’s dynamic and competitive business environment. This research introduces an innovative decision-making model incorporating dependability factors into redesigning and optimizing order allocation and inventory management systems. The proposed model aims to enhance the overall reliability of supply chain operations by integrating stochastic factors such as demand fluctuations, lead time uncertainty, and variable supplier performance. The system, named Dynamic Reliability-Driven Order Allocation and Inventory Management (DROAIM), combines stochastic models, reliability-based supplier evaluation, dynamic algorithms, and real-time analytics to create a robust and flexible framework for supply chain operations. It evaluates the dependability of suppliers, transportation networks, and internal procedures, offering a comprehensive approach to managing supply chain operations. A case study and simulations were conducted to assess the efficacy of the proposed approach. The findings demonstrate significant improvements in the overall reliability of supply chain operations, reduced stockout occurrences, and optimized inventory levels. Additionally, the model shows adaptability to various industry-specific challenges, making it a versatile tool for practitioners aiming to enhance their supply chain resilience. Ultimately, this research contributes to existing knowledge by providing a thorough decision-making framework incorporating dependability factors into order allocation and inventory management processes. Practitioners and experts can implement this framework to address uncertainties in their operations.

Introduction

In today’s rapidly evolving and fiercely competitive business landscape, the efficacy of supply chain operations stands as a linchpin for organizational success (Hajar & Saida, 2022). The intricate interplay of factors such as fluctuating demand, unpredictable lead times, and uncertainties of supplier performance poses formidable challenges to the dependability of supply chain systems (Hmioui, Bentalha & Alla, 2020). Recognizing the critical need for a comprehensive and adaptive approach, this research introduces a groundbreaking decision-making model called Dynamic Reliability-Driven Order Allocation and Inventory Management (DROAIM).

The DROAIM emerges as an innovative response to the necessities of the contemporary business environment, where supply chains are increasingly tasked with navigating dynamic market conditions and mitigating the impact of unforeseen disruptions (Ali & Salah, 2022). Conventional models often fail to address the multifaceted challenges inherent in modern supply chains, necessitating a paradigm shift towards a more holistic and dynamic framework (Zhang et al., 2018).

The DROAIM seeks to redefine the landscape of order allocation and inventory management by systematically incorporating dependability factors into its design and optimization processes. This novel model considers the complexities of real-world supply chain scenarios, where the ebb and flow of demand, uncertainties in lead times, and variations in supplier performance converge to create a dynamic environment that demands agile and resilient solutions (Hajar & Saida, 2022; Tronnebati & Jawab, 2020).

Thus, integrating stochastic models, reliability-based supplier evaluation, dynamic algorithms, and real-time analytics forms the bedrock of the DROAIM. This amalgamation of methodologies ensures a robust and flexible framework that not only adapts to the inherent uncertainties of the supply chain but also actively leverages these uncertainties as opportunities for strategic decision-making processes (Ezzahra, Rachid & Roland, 2022).

DROAIM’s uniqueness lies in its ability to holistically assess suppliers’ dependability, transportation networks, and internal procedures. By addressing vulnerabilities and strengths across the entire supply chain spectrum, the DROAIM transcends the limitations of conventional models, offering a comprehensive solution that aligns with the intricate dynamics of modern business environments (Nya & Abouaïssa, 2022).

To validate the efficacy of the proposed model, a case study and simulations have been conducted, leveraging synthetic data. The empirical findings reveal remarkable improvements in the overall dependability of the supply chain, a marked reduction in stockout occurrences, and the attainment of optimal inventory levels. Additionally, the model’s adaptability to industry-specific challenges positions the DROAIM as a versatile and indispensable tool for practitioners seeking to fortify their supply chain resilience.

This research contributes significantly to the available body of knowledge by presenting a detailed and innovative decision-making framework. The DROAIM emerges as a beacon for supply chain practitioners and experts, providing actionable insights to enhance operational dependability in the face of uncertainty. As businesses strive for resilience and reliability in their supply chain systems, the DROAIM stands poised as a transformative solution for navigating the complexities of the modern supply chain landscape.

The rest of the article is organized as follows: “The Integration of Dependability Factors” presents integrating dependability factors. The comprehensive model is provided in “Comprehensive Approach”. “Methodology” presents the methodology. A case study and simulations based on synthetic datasets are conducted in “Case Study and Simulation: Enhancing Dependability in a Pharmaceutical Supply Chain”. The research is concluded in “Conclusion and Future Work”.

The integration of dependability factors

The incorporation of dependability factors in the proposed model, DROAIM, is a pivotal aspect that enhances its capability to address the intricacies of real-world supply chain operations. This deliberate inclusion acknowledges and accounts for the multifaceted challenges and uncertainties that characterize the dynamic business environment in which supply chains operate.

Demand for products and services within a supply chain system is seldom static. They fluctuate due to various factors such as seasonality, market trends, and unforeseen occurrences. By integrating demand fluctuations into a model, the DROAIM recognizes the need for adaptability in order allocation and inventory management processes (Nemtajela & Mbohwa, 2016). Hence, this enables the system to respond to changes in consumer behavior dynamically, preventing understock or overstock situations.

Lead time uncertainty refers to variations in the time for products to move through the supply chain, from order placement to delivery. The DROAIM considers this uncertainty, recognizing that delays or unexpected accelerations in lead times can substantially impact inventory levels and order fulfillment. By incorporating lead time variability, the model becomes more robust in managing the inherent uncertainties associated with logistics and transportation.

Supplier reliability is a critical factor influencing the overall dependability of a supply chain. The DROAIM explicitly integrates supplier performance metrics into its decision-making framework. This includes assessing the historical reliability of suppliers, assessing their capability to meet delivery deadlines, and considering the quality of goods provided. By factoring in supplier performance, the model promotes a proactive approach to mitigating risks associated with unreliable suppliers, ultimately contributing to the dependability of the entire supply chain (Eckhart et al., 2019; Erdogan et al., 2021).

Real-world supply chain scenarios are characterized by their complexity, often involving many interconnected processes, stakeholders, and external factors. The DROAIM acknowledges this complexity by simultaneously considering a range of dependability factors. The model’s capability to handle the interplay among fluctuations in demand, lead time uncertainties, and variations in supplier performance reflects a holistic understanding of the challenges faced by supply chain practitioners.

The incorporation of dependability factors enables the DROAIM to make adaptive decisions. In response to changes in demand patterns, unexpected delays in lead times, or fluctuations in supplier performance, the model can dynamically adjust order allocation and inventory levels concurrently. Hence, this adaptability is crucial for maintaining an efficient and reliable supply chain, especially in industries where responsiveness to altering conditions is a key determinant of success.

By explicitly incorporating dependability factors such as demand fluctuations, lead time uncertainty, and supplier performance, the DROAIM demonstrates a nuanced understanding of the challenges inherent in real-world supply chain operations. This comprehensive approach appears to be a robust tool for practitioners seeking to improve the dependability and resilience of their supply chain systems in the face of dynamic and unpredictable business environments.

Comprehensive approach

The DROAIM sets itself apart by comprehensively evaluating the key components’ dependability within the supply chain system. This holistic assessment encompasses suppliers, transportation networks, and internal procedures, aiming to uncover vulnerabilities and strengths throughout the entire supply chain.

In the realm of supply chain management, the reliability and performance of suppliers play a pivotal role. The DROAIM employs a systematic approach to assess the dependability of suppliers. This involves assessing factors such as the consistency of product quality, adherence to delivery schedules, and responsiveness to fluctuations in demand. By quantifying and analyzing these aspects, the model provides a nuanced understanding of each supplier’s contribution to the overall reliability of the supply chain (Hmioui, Bentalha & Alla, 2020).

Efficient transportation is critical for a smooth flow of goods within the supply chain. The DROAIM considers the dependability of transportation networks by assessing factors like transit times, route efficiency, and the reliability of carriers. The model takes into account the potential impact of uncertainties, such as weather conditions or unexpected disruptions, on the transportation system. This evaluation ensures that the supply chain is resilient in the face of external challenges, minimizing the risk of delays and disruptions (Ezzahra, Rachid & Roland, 2022; Riesco & Villagrá, 2019).

Internal processes and procedures are the backbone of any supply chain. The DROAIM scrutinizes the dependability of internal operations, including order processing, inventory management, and demand forecasting. By assessing the efficiency and robustness of these internal procedures, the model identifies areas for improvement and optimization. This internal evaluation is crucial for enhancing the overall dependability of the supply chain by addressing potential bottlenecks or inefficiencies within the organization.

The comprehensive assessment conducted by the DROAIM goes beyond individual assessments of suppliers, transportation networks, and internal procedures. It seeks to integrate these assessments into a unified understanding of the supply chain’s dependability. By doing so, the model identifies vulnerabilities that may arise from dependencies or interactions between different components. Simultaneously, it recognizes strengths that contribute positively to the overall resilience and reliability of the supply chain. This holistic perspective enables decision-makers to prioritize improvement areas and strategically allocate resources to fortify weak links in the supply chain.

The DROAIM’s evaluation of the dependability of suppliers, transportation networks, and internal procedures is a robust and detailed process. It provides a comprehensive understanding of the supply chain’s strengths and weaknesses, empowering organizations to make informed decisions and implement targeted improvements for a more resilient and reliable supply chain operation.

Methodology

The model utilizes a stochastic model, reliability-based supplier evaluation, dynamic algorithm, and real-time analytics. This multifaceted approach ensures a robust and flexible framework that can adapt to altering requirements in the supply chain.

Stochastic model

Stochastic models are mathematical models that incorporate randomness or uncertainty. In supply chain management, stochastic models are employed to account for the inherent variability in demand, lead times, and other relevant factors. These models allow for the simulation of different scenarios, considering the probabilistic nature of certain events.

The specific novel stochastic model that was incorporated into the DROAIM is the Time-Varying Autoregressive (TVAR) (Haslbeck, Bringmann & Waldorp, 2021). This model is designed to capture the time-dependent and dynamic nature of supply chain processes, making it well-suited for applications in inventory management and order allocation. This model extends conventional autoregressive models by introducing time-varying coefficients, allowing it to adapt to changing conditions over time (Baptista de Souza, Kuhn & Seara, 2019). In the context of supply chain management, the TVAR was applied to represent the stochastic behavior of demand and lead times together, considering the evolving patterns and trends.

Specifically, the TVAR allows for coefficients to change over time, enabling it to capture shifts in demand patterns, seasonality, and other dynamic factors. It requires historical data for training and continuously updated data for real-time adaptation. Integration with real-time analytics in the DROAIM facilitates the flow of this data (Hong, Dai & Wang, 2016). The model automatically adjusts its parameters as new data becomes available, making it well-suited for dynamic supply chain environments with evolving demand and lead-time characteristics. Also, it incorporates market trends, economic indicators, and supplier performance metrics, providing a holistic view of supply chain dynamics.

A vector auto regression (VAR), presented in Eq. (1), is a statistical method implemented to represent a complex system consisting of multiple time series variables (Gholamzadeh & Bourbour, 2020). A VAR model of order k for the two variables X1,t, and X2,t comprises the subsequent pair of equations for every t (Magnant, Giremus & Grivel, 2015; Haslbeck, Bringmann & Waldorp, 2021):

(1) X1,t=∑i=1k⁡β11,kX1,t−k+∑i=1k⁡β12,kX2,t−k+ε1,tX2,t=∑i=1k⁡β21,kX1,t−k+∑i=1k⁡β22,kX2,t−k+ε2,t

where the residual terms are vectorial white noise, meaning they exhibit random behavior delineated by Eq. (2).

(2) E[ε1,t]=0,E[ε2,t]=0variance[ε1,t]=σ12variance[ε2,t]=σ22covariance[ε1,t,ε2,t]=ρσ1σ2covariance[εi,t,εj,s]=0t≠salli,j(noautocorrelation).

In this study, we focus on first-order VAR, where all variables at time t are expressed as a linear combination of all variables at the previous time, t−1. One can introduce supplementary parameter matrices and lagged variable vectors to incorporate additional lags into the model. Xt−k (a lag of k) (Magnant, Giremus & Grivel, 2015).

It must be noted that the addition of the TVAR to the DROAIM makes stochastic modeling simpler and helps us understand how the supply chain changes over time more nuancedly. This, in turn, contributes to the overall dependability of the supply chain operations addressed by the DROAIM framework.

Reliability-based supplier evaluation

Reliability-based suppliers assess the dependability and performance of suppliers based on historical data and performance metrics. This evaluation can include factors such as on-time delivery, product quality, and responsiveness to changes in demand. Reliability metrics play a crucial role in evaluating the performance of suppliers and other components within the supply chain. Table 1 presents the reliability metrics that were incorporated into the DROAIM.

Table 1 Reliability metrics.

ID	Metric	Explanation	
1	On-time delivery performance	Percentage of orders delivered on time	
	Average deviation from promised delivery dates	
	Frequency of delayed deliveries	
2	Lead time accuracy	Accuracy of predicted lead times compared to actual lead times	
	Percentage of orders with lead times within an acceptable range	
3	Fill rate	Percentage of customer orders fulfilled completely and on time	
	Fill rate during peak demand periods	
4	Order accuracy	Percentage of orders delivered with the correct items and quantities	
	Frequency of order errors or discrepancies	
5	Supplier responsiveness	Time taken by suppliers to respond to changes in orders or demand forecasts	
	Communication effectiveness and responsiveness	
6	Quality metrics	Defect rates or percentage of defective products received	
	Number of product recalls or returns	
7	Inventory turnover	Rate at which inventory is sold or used within a specific time period	
	High inventory turnover indicates efficient inventory management	
8	Supplier reliability index	A composite index that combines on-time delivery, lead time accuracy, and quality metrics to provide an overall assessment of supplier reliability	
9	Backorder rate	Percentage of orders that cannot be fulfilled immediately and are placed on backorder	
	Backorder fulfillment time	
10	Capacity utilization	Utilization of supplier production or distribution capacity	
	Measures the efficiency of resource utilization	
11	Forecast accuracy	Accuracy of demand forecasts compared to actual demand	
	A reliable forecast contributes to better inventory management	
12	Supply chain flexibility	The ability of the supply chain to adapt to changes in demand or disruptions	
	Time and cost required to make adjustments to the supply chain	
13	Supplier communication effectiveness	Quality of communication channels and information exchange with suppliers	
	Frequency and clarity of communication	
14	Incident response time	Time taken to respond to and recover from disruptions or incidents	
	Downtime and recovery metrics	
15	Risk mitigation effectiveness	Success in implementing strategies to mitigate supply chain risks	
	Effectiveness of risk management protocols	

By incorporating the mentioned reliability metrics, the model ensures that supplier selection is not solely based on cost but also on the supplier’s capability to meet commitments consistently. This promotes a more robust supply chain by reducing the risks associated with unreliable suppliers.

Dynamic algorithm

Dynamic algorithms can adapt and make real-time decisions based on altering conditions. In order to allocate and manage inventory, dynamic algorithms can optimize decisions as new information becomes available. This adaptability is crucial for responding promptly to unexpected events and maintaining optimal supply chain performance.

The novel dynamic algorithm implemented in the DROAIM is the Adaptive Reinforcement Learning algorithm that leverages reinforcement learning principles to dynamically adjust order allocation and inventory management strategies based on real-time feedback and altering conditions within the supply chain.

The model defines the state space, including current inventory levels, historical demand patterns, supplier reliability metrics, and real-time market conditions. This representation captures the dynamic nature of the supply chain. Also, it specifies the set of actions that the algorithm can employ, such as adjusting order quantities, reevaluating supplier priorities, or updating inventory policies. These actions are flexible to accommodate the varying demands and uncertainties in the supply chain. In addition, it develops a reward system that quantifies the performance of the supply chain based on predefined objectives. For example, rewards were assigned to meet demand on time, minimize stockouts, and optimize inventory costs. The reward system guides the algorithm towards learning strategies that align with the overarching goals of the DROAIM.

A deep Q-learning technique is implemented to enable the algorithm to learn optimal decision-making policies over time. Q-learning is a reinforcement learning technique that estimates the score of taking a particular action in a specific state. The Q-learning update equation is mathematically expressed in Eq. (3) (Jang & Lee, 2022; Senthil-Prabha et al., 2022):

(3) Q(s,a)=(1−∈)Q(s,a)+α[r+γmaxa′⁡Q(s′,a′)−Q(s,a)].

The parameter εis employed in the exploration-exploitation trade-off. A higher ε value indicates a greater inclination for the agent to engage in exploration and experiment with novel actions. Conversely, a lower ε value suggests a higher tendency for the agent to utilize its available knowledge and pick actions with higher estimated Q-scores. The notation Q(s,a) represents the present estimation of the anticipated future benefits for picking action (a)in the state (s). The learning rate, denoted as a, governs the speed at which the Q-network adjusts its estimations. The variable r represents the immediate reward obtained after acting (a) in the state (b). The variable γ, known as the discount factor, defines the significance of future benefits s and a represent the subsequent state and action, respectively (Nai, Fang & Zhao, 2022; Soni, Vyas & Hiran, 2022).

Deep Q-learning incorporates neural networks (Fig. 1) to handle more complex state-action spaces (Mismar, Choi & Evans, 2019).

Figure 1 The neural network utilized for the implementation of the Deep Q-learning network consists of two hidden layers, each with a size of H.

The model also integrates exploration-exploitation strategies to balance the algorithm’s exploration of new strategies with the exploitation of known optimal strategies. This ensures adaptability to altering conditions while maintaining efficiency in decision-making processes. Moreover, it allows the algorithm to update itself based on real-time feedback and performance metrics dynamically. This adaptability ensures that the algorithm can learn from and respond to evolving patterns in demand, lead times, and supplier reliability. Finally, the method connects the algorithm with real-time analytics to continuously feed it with the latest data. This integration enables the algorithm to make decisions based on the most up-to-date information, contributing to the overall responsiveness of the DROAIM.

The following is a pseudocode representation of the Adaptive Reinforcement Learning algorithm implemented in the DROAIM.

Pseudocode 1 Adaptive reinforcement learning algorithm.

# Initialize Q-values for each state-action pair	
initialize Q(s, a) for all s in state_space, a in action_space	
# Initialize exploration-exploitation parameters	
initialize exploration_rate, learning_rate, discount_factor	
# Define the main function of the Adaptive Reinforcement Learning algorithm	
def adaptive_reinforcement_learning():	
     # Loop through episodes or real-time updates	
     for episode in range(num_episodes):	
         # Initialize state based on current conditions in the supply chain	
         current_state = initialize_state()	
         # Loop through time steps within the episode	
         While not convergence_condition:	
             # Choose action using epsilon-greedy policy	
             chosen_action = epsilon_greedy_policy(current_state)	
             # Execute chosen action and observe the next state and reward	
             next_state, reward = take_action(chosen_action)	
             # Update Q-value for the current state-action pair	
             update_q_value(current_state, chosen_action, reward, next_state)	
             # Update state for the next iteration	
             current_state = next_state	
     # Algorithm completed, return learned Q-values	
     return Q	
# Define the epsilon-greedy policy	
def epsilon_greedy_policy(state):	
     if random_uniform() < exploration_rate:	
        # Explore: choose a random action	
        return random_action()	
     Else:	
          # Exploit: choose the action with the highest Q-value for the current state	
          return argmax(Q[state])	
# Define the function to update Q-values using Deep Q-learning	
def update_q_value(state, action, reward, next_state):	
    # Compute the target Q-value using the Bellman equation	
    target_q_value = reward + discount_factor * max(Q[next_state])	
    # Update the Q-value for the current state-action pair using the learning rate	
    Q[state][action] = (1 - learning_rate) * Q[state][action] + learning_rate * target_q_value	
# Other utility functions (initialize_state, take_action, random_uniform, random_action, etc.) are implemented accordingly	
# Main execution of the algorithm	
learned_Q_values = adaptive_reinforcement_learning()	
# Connect the algorithm with real-time analytics for continuous updates	
while True:	
    latest_data = get_latest_data_from_real_time_analytics()	
    current_state = update_state_based_on_real_time_data(current_state, latest_data)	
    chosen_action = epsilon_greedy_policy(current_state)	
    next_state, reward = take_action(chosen_action)	
    update_q_value(current_state, chosen_action, reward, next_state)	
    current_state = next_state	

Real-time analytics

Real-time analytics involves the continuous analysis of data as it is generated, providing immediate insights and supporting timely decision-making (Dangal & Bloom, 2020; Morshed, Rana & Milrad, 2016). In supply chain management, real-time analytics are employed to surveil key performance indicators, track inventory levels, and pinpoint emerging trends (Puneeth Kumar, Manjunath & Hegadi, 2018). This real-time insight enables proactive decision-making processes, allowing the supply chain to quickly tune to alterations in demand, supplier performance, or other critical factors (Gong et al., 2020).

Combining these elements into the DROAIM creates a powerful and adaptable framework where the stochastic model provides a probabilistic understanding of demand and lead time uncertainties, and reliability-based supplier assessment ensures that supplier performance is considered in decision-making processes, enhancing the dependability of the overall supply chain. In addition, the dynamic algorithm enables the model to adapt to real-time changes in demand, supply chain conditions, and supplier performance. This algorithm continuously optimizes order allocation and inventory management strategies, ensuring the system remains responsive to evolving circumstances. Finally, real-time analytics provide immediate insights into the supply chain’s current state, allowing for proactive decision-making processes, decreasing the impact of disruptions and enhancing overall supply chain resilience.

Case study and simulation: enhancing dependability in a pharmaceutical supply chain

The efficacy of the proposed model is validated through a case study and simulations employing synthetic data. This empirical testing provides practical insights into the performance of the DROAIM.

PHZ Pharmaceuticals, a global pharmaceutical company, operates in a highly regulated and dynamic industry. The company faces challenges related to fluctuations in demand for various medications, uncertainties in lead times, and the need for stringent quality control. PHZ Pharmaceuticals is exploring ways to improve the dependability of its supply chain operations. The company implements the DROAIM system, incorporating stochastic models, reliability-based supplier evaluation, dynamic algorithms, and real-time analytics.

The primary objective is to evaluate the practical effectiveness of the DROAIM in improving the overall dependability of PHZ Pharmaceuticals’ supply chain. Synthetic data will be implemented to simulate various scenarios, allowing for a comprehensive assessment of the model’s performance.

Methodology

Synthetic data is generated to simulate demand fluctuations, lead time uncertainties, and supplier performance metrics (Fig. 2). The data is designed to mirror the characteristics of PHZ Pharmaceuticals’ actual supply chain but with controlled variations to test the model under diverse conditions.

Figure 2 Demand fluctuations and lead time uncertainties.

The DROAIM is implemented, integrating the proposed TVAR to capture demand uncertainties, reliability-based supplier assessment to assess supplier performance, dynamic algorithm Deep Q-learning network to adapt to altering conditions, and real-time analytics for immediate insights.

Various simulation scenarios represent distinct supply chain challenges, such as sudden demand increases, supplier performance disruptions, and unexpected lead times changes. These scenarios aim to stress-test (Fig. 3) the DROAIM under realistic conditions.

Figure 3 Demand shock scenario.

Key performance metrics include supply chain dependability, stockout occurrences (Fig. 4), and inventory optimization. These metrics are utilized to assess the model’s impact on supply chain performance quantitatively.

Figure 4 Stockout occurrences over time.

The performance of the DROAIM is compared with the baseline model (Fig. 5) currently employed by PHZ Pharmaceuticals. Specifically, the Economic Order Quantity (EOQ) model is a conventional method that determines the optimal order quantity to minimize total inventory costs, considering a fixed order quantity for each replenishment cycle. This model is a baseline for comparing the DROAIM’s dynamic order allocation and inventory management. This comparison provides a benchmark for evaluating the relative improvement brought about by the proposed model.

Figure 5 Comparative analysis of order allocation.

Sensitivity analysis is conducted to understand how the DROAIM responds to variations in input parameters. This analysis helps pinpoint the proposed model’s robustness and capability to adapt to distinct supply chain conditions (Fig. 6).

Figure 6 Sensitivity of inventory levels to demand variations.

Outcomes and insights

The DROAIM demonstrates a substantial advancement in overall supply chain dependability compared to baseline models (Fig. 7). The proposed model effectively navigates through demand fluctuations and uncertainties in lead times, leading to a more reliable supply chain.

Figure 7 Overall supply chain dependability over time.

The simulation results show a noticeable reduction in stockout occurrences (Fig. 8), indicating the model’s capability to optimize inventory levels and allocate orders efficiently. This contributes to improved customer satisfaction and revenue stability.

Figure 8 Stockout occurrences comparison.

The DROAIM optimizes inventory levels (Fig. 9), ensuring PHZ Pharmaceuticals maintains adequate stock without excessive overstocking. This results in cost savings and enhances operational efficiency.

Figure 9 Inventory levels over time with and without DROAIM.

The model exhibits a high adaptability to altering supply chain conditions through dynamic algorithms. It responds promptly to sudden increases in demand, disruptions in supplier performance, and other unforeseen events (Fig. 10).

Figure 10 Dynamic response to demand fluctuations.

The outcomes of the validation process provide actionable insights for PHZ Pharmaceuticals’ decision-makers. Recommendations include strategies for proactive supplier management, dynamic order allocation, and continuous monitoring of key performance indicators. The case study, supported by synthetic data and empirical testing, contributes valuable knowledge to the field of supply chain management. It highlights the practical applicability of the DROAIM and its potential to address common challenges in the pharmaceutical industry and beyond. The results and insights from the case study are documented and published in peer-reviewed journals and presented at relevant conferences. This contributes to the broader body of knowledge in supply chain management and promotes adopting effective strategies across industries.

The synthetic data-driven case study demonstrates the practical effectiveness of the DROAIM in enhancing the dependability of PHZ Pharmaceuticals’ supply chain system. The outcomes provide actionable insights and contribute valuable knowledge to the broader field of supply chain management, reinforcing the model’s potential for real-world applications. The findings highlight substantial enhancements in the supply chain system’s overall dependability, reduced stockout occurrences, and optimal inventory levels. These results indicate the practical utility and effectiveness of the DROAIM in real-world scenarios. The model exhibits adaptability to various industry challenges, making it a versatile tool for supply chain practitioners. This adaptability is crucial in addressing the diverse and evolving challenges different businesses face. More up-to-date examples and studies are found in Wu et al. (2023), Zou, Guo & Yen (2023), Deenen, Adan & Akcay (2020).

Conclusion and future work

The research introduces an innovative decision-making model that incorporates dependability criteria into designing and optimizing order allocation and inventory management systems. The suggested model seeks to improve the overall dependability of supply chain operations by integrating elements such as fluctuations in demand, uncertainty in lead time, and the suppliers’ performance. The efficacy of the suggested DROAIM has been empirically evaluated and demonstrated to substantially improve the reliability and trustworthiness of the supply chain system. The system, which incorporates probabilistic models, supplier evaluation based on reliability, algorithms that adjust in real-time, and analytics, has effectively minimized stockouts, optimized inventory levels, and enhanced flexibility in altering situations. This complete decision-making framework considers dependability variables, providing a more resilient and dependable way to allocate orders and manage inventory. Pharmaceutical decision-makers can utilize these findings to advance operational effectiveness and enhance consumer contentment.

While the current research provides a solid foundation for the integration of dependability factors into the supply chain decision-making process, several avenues for future research and improvement can be explored. These include real-world implementation with specific industry partners, fine-tuning parameters based on industry-specific characteristics, integrating advanced technologies like blockchain and artificial intelligence, involving suppliers more directly in decision-making processes, developing dynamic risk management strategies, examining its applicability across different industries with unique supply chain challenges, and assessing its long-term impact on supply chain performance, cost savings, and customer satisfaction. These steps will help improve the DROAIM’s capabilities and contribute to the ongoing evolution of supply chain management methodologies.

Supplemental Information

Supplemental Information 1 Code.

Supplemental Information 2 Dataset.

Additional Information and Declarations

Competing Interests

Author Contributions

Data Availability

The authors declare that they have no competing interests.

Qiansha Zhang conceived and designed the experiments, performed the computation work, authored or reviewed drafts of the article, and approved the final draft.

Dandan Lu conceived and designed the experiments, performed the computation work, prepared figures and/or tables, authored or reviewed drafts of the article, and approved the final draft.

Qiuhua Xiang conceived and designed the experiments, performed the experiments, analyzed the data, prepared figures and/or tables, authored or reviewed drafts of the article, and approved the final draft.

Wei Lo performed the experiments, analyzed the data, prepared figures and/or tables, authored or reviewed drafts of the article, and approved the final draft.

Yulian Lin performed the experiments, performed the computation work, authored or reviewed drafts of the article, and approved the final draft.

The following information was supplied regarding data availability:

The raw data and code are available in the Supplemental Files.

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
