# Peer review of "Design and optimization of dynamic reliability-driven order allocation and inventory management decision model"

_PeerJ Computer Science, doi:10.7717/peerj-cs.2294_

## Round 0.1 · original submission · Major Revisions

Dear Dr. Lu

Your manuscript has been reviewed with interest by the relevant experts in the field. based on their input below, you will see that the manuscript requires a number of changes to be incorporated. please carefully revise the article and resubmit, please also respond the following queries in addition to the reviewer comments

My Editor Comments:

Please write a clearer statement of the specific problems currently faced in the industry that the new model aims to solve

Describe the sequence of steps in the model, from data collection to decision-making, to give readers a clear understanding of the methodology in the form of a flow chart

Please provide more details on the case study, such as the industry it pertains to etc.

Break down complex sentences into simpler ones for better readability. Ensure that the abstract flows logically from one point to the next.

Reviewer 1 ·

Basic reporting

The findings of This paper indicate substantial enhancements in the overall dependability of the supply chain, decreased occurrences of stockouts, and optimal levels of inventory. Moreover, the model displays adaptability to numerous industry-specific difficulties, making it a versatile tool for practitioners trying to strengthen their supply chain resilience.
I have the following Comments and Suggestions for further Improvement of the paper.

The paper identifies the importance of efficient order allocation and inventory management in supply chain operations, which is highly relevant in today's competitive business environment.
Consider providing specific examples or case studies from the current industry landscape to highlight further the relevance and urgency of addressing these challenges.
The introduction of the Dynamic Reliability-Driven Order Allocation and Inventory Management (DROAIM) model is innovative and well-thought-out, incorporating stochastic models, reliability-based evaluations, and real-time analytics.

Experimental design

Further detail on how DROAIM differentiates from existing models and frameworks in the literature would enhance the understanding of its unique contributions.

Providing a step-by-step breakdown of the methodology with flowcharts or diagrams can help readers better grasp the implementation process of the DROAIM model. The case study uses synthetic data, incorporating real-world data and comparing results could significantly strengthen the credibility and applicability of the findings.

Validity of the findings

A more detailed analysis of the results, including statistical significance and comparison with other models, would add depth to the evaluation and reinforce the robustness of the findings. Discussing potential limitations of the study and future research directions could provide a more balanced view and encourage further exploration in this area. Including feedback from industry practitioners who have tested or reviewed the model could provide additional validation and practical insights.

Reviewer 2 ·

Basic reporting

In this article, the authors have proposed an Efficient order allocation and inventory management system using stochastic models, reliability-based supplier evaluation, dynamic 23 algorithms, and real-time analytics. The study seems to be fine and contributes to the body of knowledge but it can be further strengthened by incorporating the following comments.

Experimental design

It would be better to write the numerical results of the model in the abstract so that readers can get a good idea about the contribution of the model.
Figures 9 and 10 need more elaboration about their understanding to the readers.
Please clearly write about, which is the baseline model discussed in Figure 8,
Please write the Adaptive Reinforcement Learning algorithm in a more professional and scientific way
Clearly state the specific objectives and contributions of the proposed model in the introduction. This helps readers quickly grasp the novelty and significance of your work.

Validity of the findings

The abstract is well-detailed but could benefit from a slight reduction in length for conciseness. Focus on the core contributions and results to make it more impactful.

Please explicitly state the primary problems faced in current order allocation and inventory management systems that your model addresses.

Provide a brief but clear description of the key components of the DROAIM model, such as how the stochastic models and dynamic algorithms are utilized. This will help readers understand the innovative aspects of your approach.
Emphasise what makes the DROAIM model innovative compared to existing models.
Include specific quantitative results from the simulations to provide a clearer picture of the improvements.

Additional comments

The language of the article needs improvement by a professional editor.

---

## Round 0.2 · accepted · Accept

Dear authors,

Thank you for re-submitting your manuscript after incorporating the comments of the experts. Based on the input received from the experts, I am pleased to notify that we are now satisfied with the current quality of the manuscript and therefore recommend it for publication. Thank you for your fine contribution.

Reviewer 1 ·

Basic reporting

The authors have carefully taken into account all the feedback from the major revision. They've made the necessary changes and improvements, ensuring that the manuscript is now stronger and more aligned with the reviewers' suggestions

Experimental design

The authors have carefully taken into account all the feedback from the major revision. They've made the necessary changes and improvements, ensuring that the manuscript is now stronger and more aligned with the reviewers' suggestions

Validity of the findings

The authors have carefully taken into account all the feedback from the major revision. They've made the necessary changes and improvements, ensuring that the manuscript is now stronger and more aligned with the reviewers' suggestions

Additional comments

The authors have carefully taken into account all the feedback from the major revision. They've made the necessary changes and improvements, ensuring that the manuscript is now stronger and more aligned with the reviewers' suggestions

Reviewer 2 ·

Basic reporting

The authors have addressed all the shortcomings. The paper can be accepted in its current state.

Experimental design

NA

Validity of the findings

NA

Additional comments

NA